# Diverse Consensuses Paired with Motion Estimation-Based Multi-Model Fitting

## Wenyu Yin
Fujian Key Laboratory of Sensing and Computing for Smart City, School of Informatics, Xiamen University
Xiamen, China
Key Laboratory of Multimedia Trusted Perception and Efficient Computing, Ministry of Education of China, Xiamen University
Xiamen, China
yinwenyu@stu.xmu.edu.cn

## Shuyuan Lin
College of Cyber Security/College of Information Science and Technology, Jinan University
Guangzhou, China
swin.shuyuan.lin@gmail.com

## Yang Lu*
Fujian Key Laboratory of Sensing and Computing for Smart City, School of Informatics, Xiamen University
Xiamen, China
Key Laboratory of Multimedia Trusted Perception and Efficient Computing, Ministry of Education of China, Xiamen University
Xiamen, China
luyang@xmu.edu.cn

## Hanzi Wang
Fujian Key Laboratory of Sensing and Computing for Smart City, School of Informatics, Xiamen University
Xiamen, China
Key Laboratory of Multimedia Trusted Perception and Efficient Computing, Ministry of Education of China, Xiamen University
Xiamen, China
hanzi.wang@xmu.edu.cn

## Abstract

Multi-model fitting aims to robustly estimate the parameters of various model instances in data contaminated by noise and outliers. Most previous works employ only a single type of consensus or implicit fusion model to represent the correlation between data points and model hypotheses. This approach often results in unrealistic and incorrect model fitting in the presence of noise and uncertainty. In this paper, we propose a novel method of diverse Consensuses paired with Motion estimation-based multi-Model Fitting (CMMF), which leverages three types of diverse consensuses along with inter-model collaboration to enhance the effectiveness of multi-model fusion. We design a Tangent Consensus Residual Reconstruction (TCRR) module to capture motion structure information of two points at the pixel level. Additionally, we introduce a Cross Consensus Affinity (CCA) framework to strengthen the correlation between data points and model hypotheses. To address the challenge of multi-body motion estimation, we propose a Nested Consensus Clustering (NCC) strategy, which formulates multi-model fitting as a motion estimation problem. It explicitly establishes motion collaboration between models and ensures that multiple models are well-fitted. Extensive quantitative and qualitative experiments are

conducted on four public datasets (i.e., AdelaideRMF-F, Hopkins155, KITTI, MTPV62), and the results demonstrate that our proposed method outperforms several state-of-the-art methods.

## CCS Concepts

• **Computing methodologies** → *Motion capture*; *Activity recognition and understanding*.

## Keywords

Multi-model fitting; Motion estimation; Tangent consensus; Cross Consensu; Nested consensus

**ACM Reference Format:**
Wenyu Yin, Shuyuan Lin, Yang Lu, and Hanzi Wang. 2024. Diverse Consensuses Paired with Motion Estimation-Based Multi-Model Fitting. In *Proceedings of the 32nd ACM International Conference on Multimedia (MM '24), October 28-November 1, 2024, Melbourne, VIC, Australia.* ACM, New York, NY, USA, 10 pages. https://doi.org/10.1145/3664647.3681646

*Corresponding author: Yang Lu (luyang@xmu.edu.cn)

## 1 Introduction

Multi-model fitting is a fundamental but challenging computer vision task with many potential applications, including image matching [9, 22], motion segmentation [1, 19], image stitching [21, 24], point cloud registration [15, 18], and geometric entity detection [14, 17]. In essence, multi-model fitting is the process of estimating the parameters of various models from a set of input data containing noise and outliers. Typically, this input data consists of feature points on object surfaces tracked across image pairs and video sequences. Therefore, the multi-model fitting problem can be reformulated as estimating moving objects across various image pairs or video sequences. In the state-of-the-art algorithms, the task of finding an unknown number of model instances is achieved

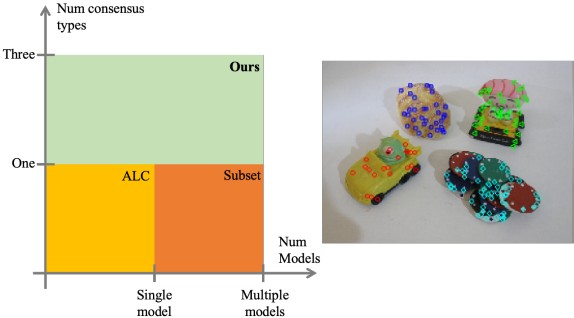

**Figure 1: Left: Differences among our CMMF method, ALC [46], and Subset [61]. ALC and Subset, which are some of the fastest SOTA methods, consider only one type of consensus, while our method achieves a more accurate clustering performance in multi-model fitting. Right: Qualitative results of CMMF on the AdelaideRMF dataset [58].**

by classifying data points into moving objects, each representing a particular model instance. Thus, the primary challenge lies in devising methodologies that can effectively estimate the parameters of multiple models and concurrently segment input data, all in the absence of prior knowledge regarding the accurate number of structures. Additionally, this task becomes more complex when input data is contaminated with both gross and pseudo outliers.

Over the past decades, the requirement to explore moving object trajectories in motion estimation has ceaselessly inspired research into robust multi-model fitting methods. Among such, subspace-based [57], optimization-based [2], and affinity-based [13] methods are three representatives. Subspace-based methods [31, 36] treat the point constraint as a labeling problem. In a subspace, each point is described as a linear combination of its corresponding points. However, these methods cannot be extended to non-linear correspondence combinations to define models. Typically, optimization-based methods [3, 4] employ a hypothesis-and-validation strategy. The hypothesis stage generates a set of candidate model hypotheses using a random or guided sampling technique. The validation stage selects model hypotheses and applies graph notation techniques to minimize an energy function. To cover all the required model instances, these methods often necessitate generating numerous model hypotheses to account for potential models. Nonetheless, it is still difficult to guarantee obtaining a model and decreases computational costs. In contrast, affinity-based methods [44, 49] can be formulated as a clustering problem. This problem is solved by projecting high-order affinity between data points onto a graph and then applying spectral clustering techniques. However, these methods are sensitive to outliers and noise. They become expensive when calculating all potential high-order affinities.

In view of these problems, we propose a novel motion estimation-based multi-model fitting method that integrates three types of diverse consensuses. The proposed method comprehensively explores the intricate relationships among data points, hypotheses, and their mutual interactions across multiple models. By redefining the problem in terms of motion estimation, we can achieve more accurate and reliable results, even in the presence of noisy

or incomplete data. Specifically, the tangent consensus residual reconstruction module rebuilds the shape and structure information at the pixel level. It applies the tangent consensus technique to identify valuable model hypotheses from residual information. This module ensures that only the relevant and accurate hypotheses are considered for further analysis. Additionally, we establish cross consensus relationships between points and hypotheses by generating a consensus affinity matrix. This matrix strengthens the correlations among inliers and reduces the sensitivity of the model to outliers. This step significantly improves the robustness of our method against noisy or incomplete data. Finally, we introduce a nested consensus clustering strategy to analyze the potential connections between multiple parameter models from a motion perspective. This strategy ensures the consistency and coherence of multiple model matrices, resulting in more accurate and reliable multi-model fitting.

The key contributions of this paper can be summarized as follows:

- We propose a novel multi-model fitting method based on motion estimation that explores three types of diverse consensuses and the potential motion interactions among multiple models. This method effectively reduces the sensitivity to noise and outlier data, improves the accuracy of model selection, and enhances the multi-model fusion capability.
- We propose a tangent consensus residual reconstruction module to optimize the correspondence between data points, which goes beyond the traditional Sampson metric by leveraging the properties of tangent consensus.
- We propose a nested consensus clustering strategy that guides the motion estimation of underlying models. This strategy combines the interaction of motion through nested consensus, which in turn improves the accuracy of parameter estimation.
- We evaluate the proposed method on four publicly available datasets. Both qualitative and quantitative results validate the applicability and superiority of our method over several state-of-the-art methods in terms of clustering accuracy and fitting performance.

## 2 Related Work

### 2.1 Analysis-Based Methods

Analysis-based model fitting methods have been proposed to examine the relationship between data points and model hypotheses. These methods can be categorized into consensus analysis-based methods [50, 65] and preference analysis-based methods [59, 67]. Consensus analysis-based methods calculate the number of inliers to match the model hypotheses and select the significant hypotheses as the model instance for estimation. The conventional method RANSAC [12] iteratively performs two processes: model hypothesis generation and model validation. In this method, the model hypotheses with the maximum consensus set are chosen as the estimated model instance, which is primarily used to process single-structure data. To address multi-structure data, sequential RANSAC [25], MultiRANSAC [68], AKSWH [56], RansaCov [41], and other representative model fitting methods are proposed to improve RANSAC. Preference analysis-based methods derive preference information to compute the residuals between data points and a set of model

hypotheses, such as RHA [66], J-Linkage [52], T-Linkage [40], RPA [42] and KF [7]. More specifically, preference information is typically utilized for data representation, which enhances the similarity matrix for data. The data are separated by combining the similarity matrix of the data with clustering methods. Subsequently, the inliers obtained from the segmentation are used to estimate the model instance. However, analysis-based model fitting methods are highly sensitive to the inlier threshold, and the accuracy of data characterization also needs to be improved. When there are a large number of outliers and multiple model structures in data, the computational efficiency of model fitting will be significantly affected. These limitations restrict their usefulness in complex scenes.

## 2.2 Deep Learning-Based Methods

Multi-model fitting methods based on deep learning have emerged in recent years, owing to their powerful learning and expressive capabilities. For instance, CONSAC [28] aims to infer search strategies directly from data. This method introduces neural network-guided model estimation based on previously estimated instances to select a subset of various metrics, which enables sequentially seeking model instances. LDA [60] formulates the multi-model fitting problem as one of learning deep feature embeddings that are clustering-friendly. In other words, this network embeds the points belonging to the same clusters together. ULRF [54] claims that robust model fitting can be effectively solved by using an unsupervised learning framework, which efficiently trains and explores search trees through the use of backbone networks. FSNet [5] generates simple matrix hypotheses and predicts the angular translation and rotation errors of image pairs. It is a framework that employs correspondences to formulate model hypotheses and integrates epipolar geometry into the attention layer. However, it does not use epipolar geometry to score the hypotheses throughout the RANSAC cycle. Moreover, these methods solve multi-model fitting problems by utilizing the priors learned from data, which constrains their interpretability. Additionally, it may also be necessary to retrain task-specific models when applying them to fit various multi-structure and multi-type geometric models.

## 2.3 Motion Estimation-Based Methods

Motion estimation-based model fitting methods utilize the motion estimation of objects in images or videos to fit models. Subspace-based methods, such as ORK [6], initially reject outliers by removing data with low vector norms from the main subspace. Then, they restore the number of clusters using the N-cut and K-means algorithm to obtain the model parameters. ELSA [64] enhances the local subspace and selects a model based on the probability density function, which calculates the number of motions using an eigenvalue spectral threshold to achieve the model fitting results. Optimization-based methods typically construct a function and optimize this function. JESS [63] offers a sparse optimization method that iteratively corrects motion segmentation and removes outliers. This method reconstructs model fitting using the desired motion structure. To address the challenge of unknown model numbers, HOM [30] designs a pseudo-boolean formulation to optimize multi-graph decomposition and construct a high-order loss function. Affinity-based methods estimate models by constructing

affinity matrices and performing clustering. MSSC [29] integrates motion information from all frames in a video sequence into a correlation matrix. This method applies spectral clustering to estimate the number of motions and fit multiple models. MCMS [23] is designed for consistent affinity segmentation across all geometric models. It obtains common structure information through block diagonals to select consistent data. However, these methods often face challenges in handling video sequences and image pairs with camera motion, occlusion, and strong perspective effects.

In this work, unlike some conventional multi-model fitting methods that only utilize a single type of consensus and implicitly fuse models, we propose a method that integrates the advantages of analysis-based and motion estimation-based methods. It owns both the comprehensive conciseness of model selection and the flexibility of clustering.

## 3 Methodology

### 3.1 Residual Reconstruction with Tangent Consensus

For multi-model fitting of two-view images and video sequences, identifying the correct point correspondences is essential for sampling the minimum subset of data and generating a set of precise model hypotheses. In model fitting, the Sampson error [39] is commonly utilized to approximate the geometric distance between a point and a model hypothesis to identify this correspondence. However, it only evaluates certain geometric quantities and fails to approximate the true reprojection error, thereby ignoring shape and motion structure information at the image level. To address this limitation, we design a tangent consensus residual reconstruction module to screen residuals and select valuable model hypotheses at the pixel level, as shown in Figure 2.

Let $P = \{p_1, p_2, ..., p_n\}$ be a set of input data containing $n$ feature points. We randomly select a minimal number of $m$ points to generate three model hypothesis matrices represented by $\mathbf{A}$, $\mathbf{H}$, and $\mathbf{F}$, which denote the affine matrix $\mathbf{A}$, the homography matrix $\mathbf{H}$, and the fundamental matrix $\mathbf{F}$, respectively. The minimum numbers of points required to generate these models are 3, 4, and 8, respectively.

Suppose that we are given a set of correspondences $C = \{p_i, p_j\} \in \mathbb{R}^{2 \times 2}$ with a few outliers, where $i$ and $j$ are the index of the data point. The residual with respect to each hypothesis and data point is derived from the traditional Sampson error:

$$\varepsilon = \frac{(p_j^\mathsf{T} \mathbf{E} \, p_i)^2}{\left| \mathbf{E} \, p_i \right|^2 + \left| \mathbf{E}^\mathsf{T} p_j \right|^2}, \tag{1}$$

where $\mathbf{E}$ is the essential matrix. This residual is approximated by the geometric error [48], which is the distance between the nearest pair of points that satisfy the epipolar constraint $I(\cdot)$:

$$I(\tau) = t_j^\mathsf{T} \mathbf{E} \, t_i, \tag{2}$$

where $\tau = \{t_i, t_j\} \in \mathbb{R}^{2 \times 2}$ represents the correct correspondence, and $t_i$ and $t_j$ correspond to $p_i$ and $p_j$, respectively.

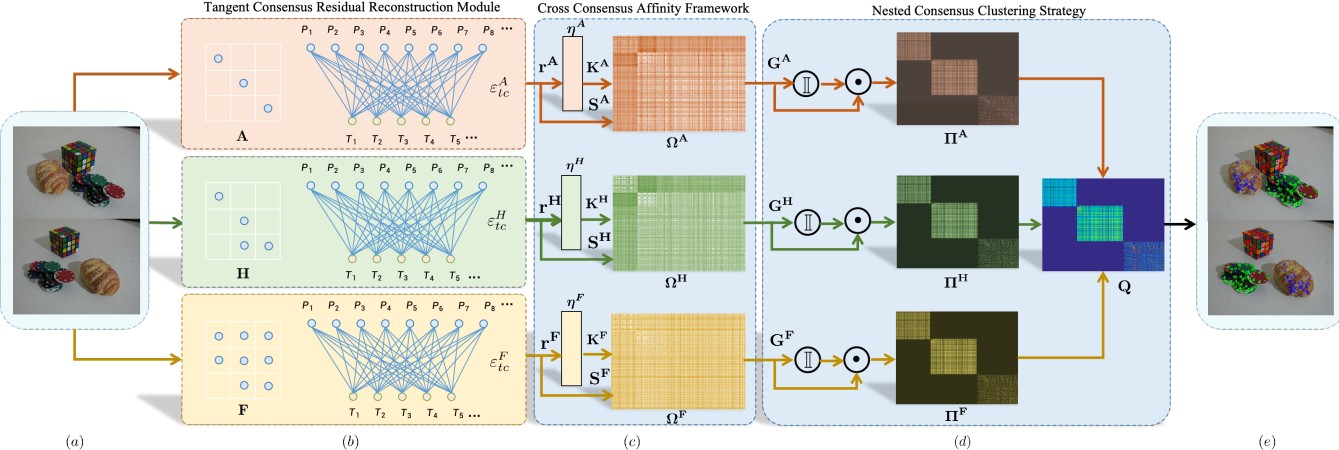

**Figure 2: Overview of the proposed CMMF method. (a) An input image pair. (b) The tangent consensus residual reconstruction $\varepsilon_{tc}$ between the points $P$ and the model hypotheses $T$. (c) The affinity matrix $\Omega$ captured by the cross consensus affinity framework. (d) The nested consensus clustering strategy merges the nested consensus sparse affinity matrices $\Pi$ from various models and groups the points into moving objects. (e) The final mutli-model fitting result. Points corresponding to different moving objects are indicated by different colors.**

We then reconstruct the residual derived from the tangent consensus error $\varepsilon_{tc}$ for matching the motion estimation as follows:

$$\varepsilon_{tc} = \min_{C} ||\tau - C||^2$$
$$s.t. \quad I(C) = \Psi(p_j)^\mathsf{T} \mathbf{E} \, \Psi(p_i), \tag{3}$$

where $\Psi(\cdot)$ is the tangent inverse function, which depends on the forward projection function $\Phi(\cdot)$. The forward projection function is adopted from the Kannala-Brandt model [26]:

$$\Phi(\tau) = \frac{\sqrt{t_i^2 + t_j^2}}{|t_i^2 + t_j^2|} (\alpha_i t_i, \alpha_j t_j) + (\beta_i, \beta_j), \tag{4}$$

where $\alpha$ and $\beta$ are the inherent parameters. The forward projection function solely defines forward projection to restore the shape of an object on the image plane. However, during iterative solving for unprojection, it fails to obtain the structural information about the object. This limitation impacts the estimation of moving objects. Note that $\Phi(\cdot)$ supports right inverses, but they only differ in lengths assigned to the unprojected bearing vectors [51]. Nevertheless, for motion estimation, a more complex model is required to capture the structural details. Considering an arbitrary central camera model, it is necessary to optimize the continuously differentiable bearing vector at the pixel level tangent planes. Therefore, the tangent inverse function is designed for $C$:

$$\Psi(C) = \frac{\partial \Phi(C)}{\partial C} \frac{J_i \times p_i + J_j \times p_j}{C \cdot (J_i \times J_j)}, \tag{5}$$

where $J_i$ and $J_j \in \mathbb{R}^{2 \times 2}$ are the Jacobians of $p_i$ and $p_j$ with respect to the epipolar constraint, and $\times$ denotes the cross product. Inspired by [47], we linearize the epipolar constraint:

$$\tau = C - \frac{I(C)}{||J||^2} J^\mathsf{T}. \tag{6}$$

Thus we substitute and minimize Eq. (3) to obtain:

$$\varepsilon_{tc} = \left\| \frac{I(C)}{||J||^2} J^\mathsf{T} \right\|^2 = \frac{(\Psi(p_j)^\mathsf{T} \mathbf{E} \Psi(p_i))^2}{\left| p_j^\mathsf{T} \mathbf{E} J_i \right|^2 + \left| p_i^\mathsf{T} \mathbf{E}^\mathsf{T} J_j \right|^2}. \tag{7}$$

The tangent consensus residual reconstruction module reflects the efficient optimization of the epipolar constraint on the differentiable tangent plane of the points. Extracting motion structure information on a pixel basis is meaningful since the original image has a tangent measure. As long as we can determine the Jacobian of the projection function in each corresponding connection, it works with arbitrary central camera models. Notably, this reconstruction module is cost-effective, as it requires computing only a single forward projection and Jacobian.

Through the tangent consensus error, we reconstruct the residual between each model hypothesis and data point. The residual vector based on the tangent consensus error is defined as follows:

$$\mathbf{r} = [\varepsilon_{tc}^1, \varepsilon_{tc}^2, ..., \varepsilon_{tc}^n]. \tag{8}$$

The reconstructed residual not only measures the geometric distance corresponding to the point but also approximates the true reprojection error. This is because the forward projection function can recover the shape information, meanwhile, the tangent inverse function can obtain the structural information. Both of these functions provide indispensable correlations for finding the affinity matrix.

### 3.2 Affinity Captured as Cross Consensus

The affinity between two features can be characterized by alignment based on the consistency of points across all hypotheses. However, the model hypotheses generated by random sampling often include a large number of irrelevant or incorrect hypotheses. This can significantly impact the performance of motion estimation and the

accuracy of the affinity matrix, especially in scenarios involving multiple models.

To address the above issues, we propose a cross consensus affinity framework to describe the affinity between points and underlying model hypotheses. First, we sort the residual $\mathbf{r}$ of the tangent consensus reconstruction in ascending order. Then, we obtain a permutation $\boldsymbol{\eta} = \{\eta_1, \eta_2, ..., \eta_n\}$, which expresses the preference relationship between model hypotheses and data points. If two points $p_i$ and $p_j$ are inliers from the same structure, they will have a lot of shared hypotheses at the top of their preference lists. Motivated by [45], we calculate the cross kernel $k$ of two points $p_i$ and $p_j$ as follows:

$$k(p_i, p_j) = \frac{1}{\rho} \sum_{x=1}^{n} \left( \left| \eta_i^{1\sim x+1} \cap \eta_j^{1\sim x+1} \right| - \left| \eta_i^{1\sim x} \cap \eta_j^{1\sim x} \right| \right), \quad (9)$$

where $\rho$ is a bandwidth, $x$ is the element number of $\boldsymbol{\eta}$, and $\eta_{(\cdot)}^{a\sim b}$ represents the set made up of elements in $\eta_{(\cdot)}$ from the $a$-th to the $b$-th. The notation $|\cdot \cap \cdot|$ indicates the cross of elements having the same indices in two permutations.

Next, for each pair of points in the three model hypothesis matrices, the cross kernel matrix $\mathbf{K}$ is computed as follows:

$$\mathbf{K}^\lambda = \begin{bmatrix} k(p_1, p_1)^\lambda & \cdots & k(p_1, p_n)^\lambda \\ \vdots & \ddots & \vdots \\ k(p_n, p_1)^\lambda & \cdots & k(p_n, p_n)^\lambda \end{bmatrix}, \quad (10)$$

where $\lambda$ is defined as the set of three model hypothesis matrices (i.e., $\mathbf{A}$, $\mathbf{H}$, and $\mathbf{F}$).

Then, the adaptive weighted density estimation [8] is utilized to assess the weighted score $s$ of a model hypothesis, which corresponds to the three model hypothesis matrices:

$$s_i^\lambda = \sum_{i=1}^{n} \frac{\mathbb{K}(r_i^\lambda)}{\kappa^\lambda \rho^\lambda}, \quad (11)$$

where $i$ is the number of model hypotheses, $\mathbb{K}$ is the Epanechnikov kernel, $\kappa$ is the inlier scale that IKOSE calculated [56], and $\rho$ is a bandwidth. The weighted scores will assign a high score to meaningful model hypotheses via Eq. (11). The cumulative weighted score for each point can be expressed as:

$$\mathbf{S}^\lambda = \sum_{j=1}^{n} s_{ij}^\lambda, \quad (12)$$

where $j$ is the number of points. The cross consensus affinity $\Omega^\lambda$ can be constructed based on the cross relationship of the points and the weighted scores of the model hypotheses:

$$\Omega^\lambda = \mathbf{S}^\lambda \mathbf{K}^\lambda = \begin{bmatrix} s_1^\lambda k(p_1, p_1)^\lambda & \cdots & s_n^\lambda k(p_1, p_n)^\lambda \\ \vdots & \ddots & \vdots \\ s_n^\lambda k(p_n, p_1)^\lambda & \cdots & s_n^\lambda k(p_n, p_n)^\lambda \end{bmatrix}. \quad (13)$$

The cross consensus affinity between two points from the same moving object should be greater, while the affinity between two points from different moving objects should be smaller. At last, an $\epsilon$−neighborhood method [29] is used to sparsify the cross consensus affinity $\Omega^\lambda$.

---

**Algorithm 1:** Diverse consensuses paired with motion estimation-based multi-model fitting (CMMF)

---

**Input:** A set of data points $P$.
**Output:** The model parameters $M$, the number of models $N$, and the data point labels $L$.

1 // Tangent Consensus Residual Reconstruction Module
2 Construct the tangent consensus residual module by Eq. (3);
3 Compute the reconstructed residual by Eqs. (7) and (8);
4 // Cross Consensus Affinity Framework
5 **for** *each type of model matrices* $\lambda \in [\mathbf{A}, \mathbf{H}, \mathbf{F}]$ **do**
6     Construct $\mathbf{K}^\lambda$ according to $\lambda$ by Eqs. (9) and (10);
7     Generate $\mathbf{S}^\lambda$ according to $\lambda$ by Eqs. (11) and (12);
8     Compute $\Omega^\lambda$ according to $\lambda$ by Eq. (13);
9 **end**
10 // Nested Consensus Clustering Strategy
11 **while** *no converged* **do**
12     **for** $\lambda \in [\mathbf{A}, \mathbf{H}, \mathbf{F}]$ **do**
13         Calculate $\mathbf{\Pi}_\lambda$ by Eq. (15);
14         $\mathbf{Q}^\lambda \leftarrow$ First eigenvectors of the Laplacian matrix;
15     **end**
16 **end**
17 $M$, $N$, and $L \leftarrow$ Cluster $\mathbf{Q}^\lambda$.

---

### 3.3 Nested Consensus Clustering Strategy

After obtaining the cross consensus affinity matrices of multiple models, the next step is to find partitions for multiple motions. We design a nested consensus clustering strategy to recover the clustering of different moving objects. Unlike previous fusion or accumulation methods, we utilize the potential relationships between different models. In the case of multi-model fitting for motion estimation, we are aware that the fundamental matrix $\mathbf{F}$ can be described as a family of $\mathbf{F} = \langle \boldsymbol{\xi} \rangle * \mathbf{H}$ parameterized by a vector $\boldsymbol{\xi}$, where $\langle \boldsymbol{\xi} \rangle$ is the skew-symmetric matrix [16], and $(*)$ represents the cross multiplication. This implies that for a given pair of points within a homography matrix $\mathbf{H}$, they are considered inliers with respect to a specific fundamental matrix. Conversely, if two points are outliers with regard to a fundamental matrix, they fail to satisfy the constraints of a homography matrix. Similarly, the homography matrix $\mathbf{H}$ can be defined as $\mathbf{H} = \lceil \zeta \rfloor \cdot \mathbf{A}$, where $\lceil \zeta \rfloor$ represents the perspective transformation, and $(\cdot)$ denotes the dot product. This means that through a series of translation and rotation transformations, the inliers of the affine matrix can be transformed into the inliers of the homography matrix, and vice versa. If a point is not an inlier of a certain homography matrix, it cannot be an inlier of the affine matrix.

Based on the potential relationships between the moving object models mentioned above, $\Omega^\mathbf{A}$, $\Omega^\mathbf{H}$, and $\Omega^\mathbf{F}$ are the sparse affinity matrices of $\mathbf{A}$, $\mathbf{H}$, and $\mathbf{F}$, respectively. We can define a nested consensus clustering strategy for the existing affinities $\Omega^\mathbf{A} \in \Omega^\mathbf{H} \in \Omega^\mathbf{F}$:

$$\min_{\mathbf{Q}} \sum_\lambda \left\{ \mathbf{Q}_\lambda^\mathsf{T} [\mathbf{\Lambda}^{(-1/2)} \Omega^\lambda \mathbf{\Lambda}^{(-1/2)}] \mathbf{Q}_\lambda \right\} - \gamma \sum_\lambda (\mathbf{Q}_\lambda^\mathsf{T} \mathbf{\Pi}_\lambda \mathbf{Q}_\lambda),$$
$$s.t. \ \mathbf{Q}^\mathsf{T} \mathbf{Q} = \mathcal{I}, \quad \mathbf{\Pi} \in \{-v, 0, v\}, \quad (14)$$

where $\mathbf{Q}$ represents the spectral embedding, $\boldsymbol{\Lambda}$ denotes the degree matrix of the sparse affinity matrix $\boldsymbol{\Omega}$, $\gamma \in \mathbb{R}$, $\mathcal{I}$ is the identity matrix, and $\boldsymbol{\Pi}$ is the nested consensus matrix. When $\pi_{ij} = v$, the nested consensus promotes a large inner product $\mathbf{Q}_{\lambda i}^{\mathsf{T}}\mathbf{Q}_{\lambda j}$, where $\mathbf{Q}_{\lambda i}$ denotes the $i$-th column. Therefore, it is desirable for points $i$ and $j$ to belong to the same cluster. There is no nested consensus for $\pi_{ij} = 0$. A different cluster assignment between $i$ and $j$ is encouraged by the nested consensus when $\pi_{ij} = -v$. This nested consensus will help affinity matrices to be further denoised or repaired.

We relax $\boldsymbol{\Pi}$ to continuous values and optimize $\boldsymbol{\Pi}$ using the affinity reconstructed from the spectral embedding $\mathbf{G} = \mathbf{Q}\mathbf{Q}^{\mathsf{T}}$. Then Eq. (14) is rewritten as:

$$\min_{\mathbf{Q}} \sum_{\lambda} \left\{ \mathbf{Q}_{\lambda}^{\mathsf{T}}[\boldsymbol{\Lambda}^{(-1/2)}\boldsymbol{\Omega}^{\lambda}\boldsymbol{\Lambda}^{(-1/2)}]\mathbf{Q}_{\lambda} \right\} - \gamma \sum_{\lambda}(\mathbf{Q}_{\lambda}^{\mathsf{T}}\boldsymbol{\Pi}_{\lambda}\mathbf{Q}_{\lambda}),$$

$$s.t.\ \mathbf{Q}^{\mathsf{T}}\mathbf{Q} = \mathcal{I},$$

$$\boldsymbol{\Pi} = \begin{cases} \mathbf{G} \odot \mathbb{I}(\mathbf{G} > 0), & \pi = 1 \\ \mathbf{G} \odot \mathbb{I}(\mathbf{G} > 0) + \mathbf{G} \odot \mathbb{I}(\mathbf{G} < 0), & \pi = 0, \\ \mathbf{G} \odot \mathbb{I}(\mathbf{G} < 0), & \pi = -1 \end{cases}$$

$$(15)$$

where $\odot$ denotes element-wise multiplication and $\mathbb{I}(\cdot)$ represents the indicator function. We assume that three model matrices are arranged as follows: the affine ($\pi = -1$) followed by the homography ($\pi = 0$) and the fundamental matrix ($\pi = 1$). According to the principle of nested consensus, $\mathbf{H}$ and $\mathbf{A}$ can be nested in $\mathbf{F}$, and $\mathbf{A}$ can be nested in $\mathbf{H}$. Finally, a separate K-means step is taken. Normalized $\mathbf{Q}$ is input to group the points into motion groups. This step helps obtain multi-model fitting results. Figure 2 depicts an overview of the proposed method. An overview of the complete steps is provided in Algorithm 1.

## 4 Experiments

### 4.1 Experimental Setup

**Datasets**. We assess the proposed method using four challenging datasets:

- **AdelaideRMF-F** [58] is a two-view multi-model fitting dataset that includes 19 pairs of images. Each image contains at least two to four objects that may be scaled, rotated, or distorted, and may also exhibit shadows or occlusions between objects. These factors contribute to the challenging nature of this dataset.
- **Hopkins155** [53] consists of two motion groups in 120 video sequences and three motion groups in 35 video sequences. It includes incomplete motion trajectories of objects, which makes motion estimation difficult.
- **KITTI** [61] comprises a series of photos captured by a camera mounted on a moving car. It contains two to five moving objects in total, including the background. The photos have a resolution of 1226 x 370 pixels. This benchmark showcases the interaction of multiple actions, intricate backgrounds, and significant camera translation.
- **MTPV62** [32] is an extension of the Hopkins155 dataset [53], involving 12 real outdoor fragments and 50 Hopkins 155 fragments. Nine of the clips exhibit strong perspective effects, making them suitable for robust perspective testing. Additionally, it provides feature trajectory information.

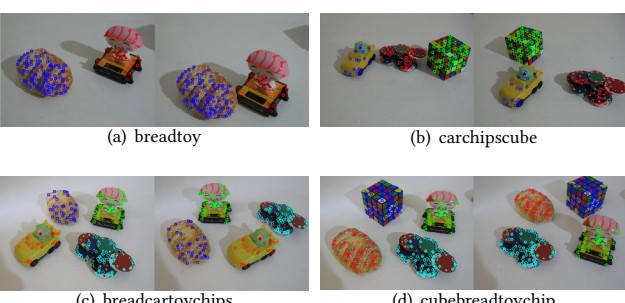

(a) breadtoy

(b) carchipscube

(c) breadcartoychips

(d) cubebreadtoychip

**Figure 3: Two-view multi-model fitting results obtained by the proposed DCMF on the AdelaideRMF-F dataset. Points corresponding to different moving objects are indicated by different colors.**

**Evaluation Metrics**. The average clustering error (ACE) and the median clustering error (MCE) are applied to evaluate the performance of multi-model fitting.

### 4.2 Comparison with State-of-the-Art Methods

**Evaluation of Two-View Model Fitting.** We first evaluate the proposed method on the AdelaideRMF-F dataset [58]. We compare the proposed method with ten state-of-the-art methods. Among these, PEARL [20], T-linkage [40], and RPA [42] are hypothesize-cluster methods. MLink [43], RansaCov [41], Prog-X [3], and LDA [60] are analysis-based methods. CBG [34], D2Fitting [33], and QUMF [11] are optimization-based methods. As shown in Table 1, the proposed method achieves the best performance in terms of ACE and MCE criteria with improvements of 0.43% and 0.89%, respectively. These improvements are attributed to the effectiveness of our tangent consensus structural details reconstruction and the multi-model nested consensus learning. Figure 3 shows some qualitative results obtained by the proposed method. We observe consistently better multi-model fitting results by our method. Specifically, for the 'breadtoy' and 'carchipscub' image pairs, our method demonstrates robustness against occlusion and shadows. Furthermore, for the 'breadcartoychips' and 'cubebreadtoychip' image pairs, our method accurately detects entire objects with multiple sub-classes. This is because our method utilizes three diverse consensuses between feature points and hypotheses, which improves the accuracy of multi-model fitting.

**Evaluation of Motion Estimation.** We further compare our method on the KITTI, Hopkins155, and MTPV62 datasets with seven state-of-the-art methods. Subspace-based single-model methods include GPCA [55], SSC [10], BDR [38], and ALC [46]. Fusion-based multi-model methods include Subset [61], MCMS [23], and HMFMS [35]. Figure 1 summarizes the core differences between them. As shown in Table 2, the subspace-based single-model methods exhibit larger ACE values. In contrast, ACE is significantly improved by the fusion-based multi-model methods. Our proposed method consistently outperforms all other competing methods in terms of ACE, with performance improvements of 1.16%, 0.12%, and 0.11%, respectively. It is noteworthy that the KITTI dataset presents various actions and intricate background effects. However, our

**Table 1: Performance comparison between the proposed CMMF and ten competing methods on the AdelaideRMF-F dataset. The best results are boldfaced.**

| Methods | PEARL | T-linkage | RPA | MLink | RansaCov | Prog-X | LDA | CBG | D2Fitting | QUMF | CMMF |
|---|---|---|---|---|---|---|---|---|---|---|---|
| ACE (%) | 29.51 | 24.12 | 17.14 | 7.53 | 15.39 | 11.04 | 9.39 | 4.51 | 4.16 | 3.85 | **3.42** |
| MCE (%) | 14.83 | 17.22 | 11.11 | 6.56 | 9.16 | 8.59 | 6.05 | 2.92 | 3.82 | 3.54 | **2.03** |

**Table 2: Performance comparison between the proposed CMMF and seven competing methods on the KITTI, Hopkins155, and MTPV62 datasets. The best results of ACE (%) are boldfaced.**

| | Methods | GPCA | SSC | BDR | ALC | Subset | MCMS | HMFMS | CMMF |
|---|---|---|---|---|---|---|---|---|---|
| KITTI | | 34.60 | 33.88 | 32.88 | 24.31 | 8.08 | 4.58 | 4.48 | **3.32** |
| Hopkins155 | Two motions | 4.59 | 1.52 | 0.95 | 2.40 | 0.23 | 0.24 | 0.21 | **0.16** |
| | Three motions | 28.66 | 4.40 | 0.85 | 6.69 | 0.58 | 0.82 | 0.67 | **0.46** |
| | All | 10.02 | 2.18 | 0.93 | 3.56 | 0.31 | 0.37 | 0.31 | **0.19** |
| MPTV62 | 12 clips | 28.77 | 17.22 | 26.63 | 0.43 | 0.30 | 0.60 | 0.55 | **0.40** |
| | 50 clips | 16.20 | 2.01 | 7.81 | 18.28 | 0.77 | 0.58 | 0.51 | **0.44** |
| | All | 16.58 | 5.17 | 5.09 | 14.88 | 0.65 | 0.58 | 0.52 | **0.41** |

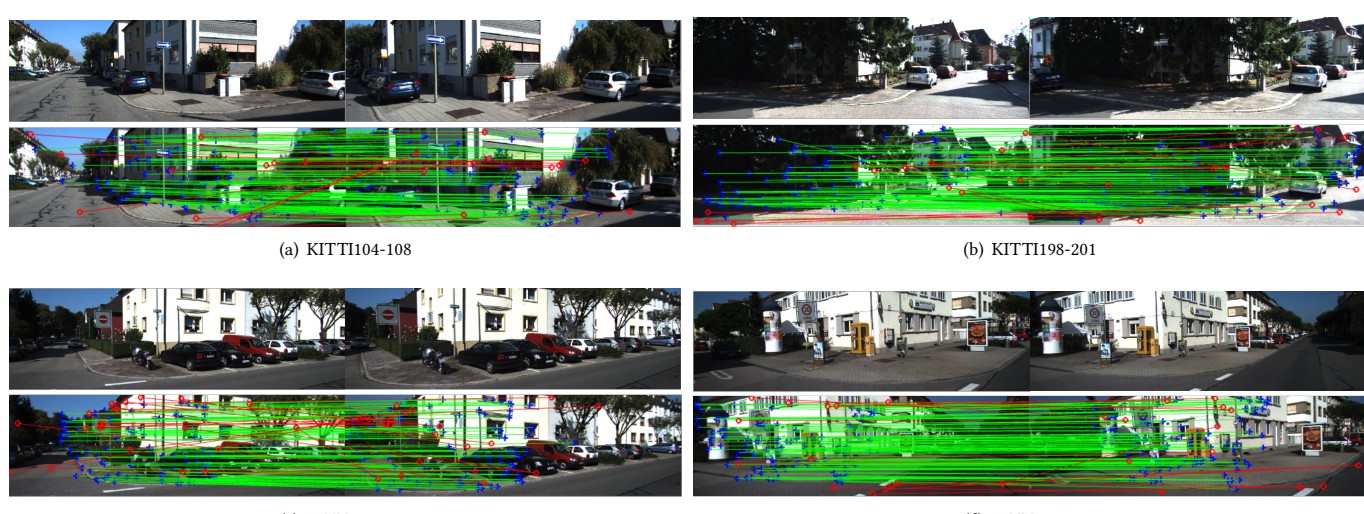

(a) KITTI104-108

(b) KITTI198-201

(c) KITTI579-582

(d) KITTI738-742

**Figure 4: Motion estimation results obtained by the proposed CMMF method from the KITTI dataset. Inlier correspondences are marked by green lines, and outlier correspondences are marked by red lines.**

proposed CMMF method effectively addresses these challenges by leveraging the nested consensus of different models. To tackle the issue of missing object motion trajectories in the Hopkins155 dataset and the strong perspective problem in the MPTV62 dataset, our method reconstructs the residual between points and hypotheses to obtain structural information of moving objects. It further enhances motion correlation between models using a cross consensus affinity framework. This method effectively preserves pixel-level motion structure information. Therefore, the proposed CMMF method offers significant performance advantages over fusion-based methods. Some qualitative estimation results obtained by our method are shown in Figure 4. In the 'KITTI104-108' and 'KITTI198-201' video sequences, the proposed method not only correctly recognizes cars

and a trash can but also properly marks walls in dark areas. In the 'KITTI579-582' and 'KITTI738-742' video sequences, our proposed method accurately labels multiple moving objects. This is because our method reduces residual-induced instability and enhances estimation performance. It achieves this by exploring the correlation of different moving objects in challenging scenarios.

### 4.3 Ablation Study

**Influence of Tangent Consensus Residual Reconstruction Module.** We demonstrate the effectiveness of the proposed tangent consensus residual reconstruction. This is achieved by approximating the true reprojection error between the points and hypotheses during the residual reconstruction of the different errors, as shown

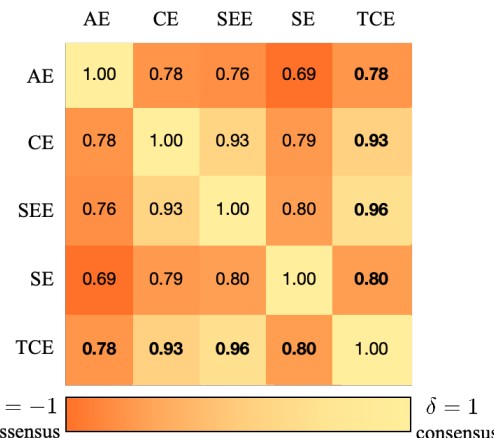

**Figure 5: Ablation study of the residual reconstruction comparing the tangent consensus error (TCE) with four competing errors. The darker the color, the worse the approximation when comparing the $\delta$ coefficients of various error metrics.**

**Table 3: Ablation study of the cross consensus affinity (CCA) framework on the MTPV62 and Hopkins155 datasets. The best results of ACE (%) are boldfaced.**

| Methods | TCRR | ORK | KA | CCA | NCC | Hopkins155 All | MTPV62 All |
|---------|------|-----|----|----|-----|----------------|------------|
| (a) | ✓ | ✓ | | | ✓ | 8.91 | 8.08 |
| (b) | ✓ | | ✓ | | ✓ | 0.36 | 0.74 |
| (c) | ✓ | | | ✓ | ✓ | **0.19** | **0.41** |

in Figure 5. We utilize Kendall's $\delta$ Rank correlation coefficient [27] to analyze a series of checkerboard-like [37] error measures. The coefficient $\delta \in [-1, 1]$ is calculated between two residual sortings. The same sorting results in $\delta$ approaching 1, and $\delta$ approaching -1 in the opposite case. Competitors are divided into two groups: one group is used for geometric errors [48], such as algebraic error (AE) and cosine error (CE), and the other group is used for image errors [47], such as symmetric epipolar error (SEE), Sampson error (SE), and tangent consensus error (TCE). We observe that the most accurate approximation of the true reprojection error for each row and column is provided by the tangent consensus error, excluding the self error. The closer $\delta$ is to 1, the greater the likelihood that two sortings come from the same structure. Our tangent consensus residual reconstruction represents a pixel-level difference in an image pair. It not only restores shape information through forward projection but also obtains structural information through inverse projection. This improves the correlation between points and hypotheses. These observations indicate that the proposed CMMF method offers an effective residual reconstruction manner for multi-model fitting to approximate the true reprojection error.

**Influence of Cross Consensus Affinity Framework.** We conduct an ablation study to explore the contribution of cross consensus affinity framework in our proposed method. As shown in Table 3, the comparison reveals that augmenting the traditional ordered residual kernels (ORK) [6] and kernels add (KA) [61] with

**Table 4: Ablation study of the nested consensus clustering (NCC) strategy on the Hopkins155, MTPV62, and KITTI datasets. The best results of ACE (%) are boldfaced.**

| Methods | Hopkins155 All | MTPV62 All | KITTI |
|---------|----------------|------------|-------|
| Affine | 0.59 | 0.82 | 15.76 |
| Homography | 0.71 | 1.08 | 11.45 |
| Fundamental | 1.79 | 3.97 | 13.92 |
| CoReg | 0.46 | 0.73 | 7.92 |
| MCMS | 0.37 | 0.58 | 4.48 |
| CMMF | **0.19** | **0.41** | **3.32** |

cross consensus notably improves results on the MTPV62 and Hopkins155 datasets. Method (a) utilizes ORK to encapsulate hypothesis affinity in a kernel. Method (b) combines KA with a superposition of multiple affinities. Our method (c) captures affinity through cross consensus. It assigns weights to different hypotheses for sampling, providing elasticity for severe sampling imbalances. This significantly boosts the performance because it is easier to reduce the impact of incorrect related information.

**Influence of Nested Consensus Clustering Strategy.** To investigate the effectiveness of the proposed nested consensus clustering strategy, we conduct extensive ablation experiments on the Hopkins155, MTPV62, and KITTI datasets. We apply the parameter $\gamma$ to promote convergence and select $\gamma = 10^{-2}$ can make the problem more easily convergent and improve the accuracy of our method. Specifically, we compare the single-model method (i.e., Affine, Homography, and Fundamental), the regularized method (CoReg) [62], and the multiplicative decomposition method (MCMS) [23]. As shown in Table 4, the proposed method consistently improves results across all methods and metrics. The nested consensus clustering strategy plays a critical role in our method. This is because most of the other competitive methods only seek one consensus spectral feature. We believe that segmenting information solely based on one consensus may not be sufficient to eliminate outliers. It may also lead to unreliable or semantically irrelevant movement. Therefore, we construct the nested consensus clustering strategy, which integrates the nested consensus of multiple models. It enhances motion connections between multiple models and improves the accuracy of multi-model fitting.

## 5 Conclusion

In this paper, we propose a highly effective method for multi-model fitting through diverse consensuses paired with motion estimation. Our method comprises three key components: the tangent consensus residual reconstruction module, the cross consensus affinity framework, and the nested consensus clustering strategy. We leverage diverse types of consensuses to analyze potential correlations between data points and model hypotheses, and we demonstrate how the proposed method can be used on potential correlations, thereby contributing to the understanding of motion collaboration between multiple models. Our method is evaluated on four challenging datasets, with the results showing its superiority over other state-of-the-art approaches. It is highly competent in practical vision tasks, excelling in fitting multiple models and achieving high clustering accuracy, even in scenarios where prior knowledge of the number of motions is lacking.

## Acknowledgments

This work was supported in part by the National Natural Science Foundation of China under Grants U21A20514, 62376233 and U22A2095; in part by the FuXiaQuan National Independent Innovation Demonstration Zone Collaborative Innovation Platform Project under Grant 3502ZCQXT2022008; in part by the Guangdong Basic and Applied Basic Research Foundation (No. 2024A1515011740); in part by the China Fundamental Research Funds for the Central Universities under Grants 20720230038, 21624404 and 23JNSYS01; in part by the Xiaomi Young Talents Program.

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
