# OpenReview forum: "Diverse consensuses paired with motion estimation-based multi-model fitting"
_acmmm.org/ACMMM/2024/Conference — MM2024 Poster_

### Official Review · Reviewer_krQq · 2024-05-17

**Rating:** 3
**Confidence:** 2

**Summary:**

The paper addresses the challenge of multi-model fitting, where traditional methods often rely on a single consensus or implicit fusion model, leading to unrealistic fitting in the presence of noise and uncertainty. The proposed method, named CMMF (Computational Multi-Model Fitting), enhances multi-model fusion by incorporating three diverse consensus mechanisms along with inter-model collaboration.
The method is evaluated using four challenging datasets (AdelaideRMF-F, Hopkins155, KITTI, MTPV62). The proposed method is compared with eight state-of-the-art methods, demonstrating the best performance in terms of ACE and MCE.

**Strengths:**

1. The introduction of a novel multi-model fitting method that explores three different types of consensuses and potential motion interactions among multiple models, reducing sensitivity to noise and outliers and improving model selection accuracy.
2. The design of a Tangent Consensus Residual Reconstruction (TCRR) module that optimizes correspondence between data points using tangent consensus, going beyond the simple Sampson metric.
3. The proposal of a Nested Consensus Clustering (NCC) strategy that guides motion estimation of underlying models, improving both estimation accuracy and parameter accuracy through motion interaction via nested consensus.
4. Extensive quantitative and qualitative experiments on four public datasets demonstrate the proposed method's superiority over state-of-the-art methods in terms of clustering accuracy and fitting performance.

**Limitations:**

1. The proposed method involves numerous iterative and matrix operation steps; however, the authors have not analyzed the computational complexity of the algorithm. It would be important to provide a theoretical analysis of the algorithmic complexity, as well as a comparison of the time cost with other algorithms.

2. How are the n input feature points extracted, and can different extraction methods affect the results of this paper? Are there different extraction methods on different datasets? This question is important because these feature points are the input part of the entire algorithm.

3. The proposed method involves the setting of multiple parameters, but the paper does not mention the process of selecting and adjusting these parameters, as well as their impact on the final results.

4. In the experiments, Evaluation Metrics used ACE and MCE. It is not explained how these two errors, ACE and MCE, are calculated. The authors should clarify why these two metrics are used and what their formulations are. Some theoretical reasoning or citations are necessary.

5. The authors have summarized Deep Learning-Based Methods in the related work section. However, the experimental part does not include a comparison with these methods.

6. The authors have compared some methods in the experimental section that are rather outdated, with some papers published a decade ago. The authors need to include comparisons with newer methods from the past few years.

**Suitability:**

2

---

### Official Review · Reviewer_ZpBn · 2024-05-25

**Rating:** 4
**Confidence:** 2

**Summary:**

The paper presents a novel approach to multi-model fitting in computer vision, which is a crucial task for applications like image matching, motion segmentation, and point cloud registration. The proposed method addresses the challenge of estimating parameters for multiple models within noisy data by introducing three key components corresponding to three consensus types. It explicitly establishes motion collaboration between models and ensures that multiple models are well-fitted. Experimental results show superior performance compared to several SOTA methods.

**Strengths:**

- Combining three types of consensus makes sense to boost the overall performance. The paper introduces a novel multi-model fitting approach that leverages diverse consensus mechanisms and motion estimation, which is an interesting idea from traditional single-consensus methods.
- Very good experimental results. By combining three different consensus mechanisms (Tangent Consensus, Cross Consensus Affinity, and Nested Consensus Clustering), the method can handle complex scenarios with multiple moving objects more effectively. The proposed TCRR module and CCA framework enhance the method's robustness against noise and outliers, which are common in real-world vision tasks.

**Limitations:**

- My main concern is about the efficiency. The proposed method involves complex calculations, which might lead to high computational overhead, making it less suitable for real-time applications or very large datasets. Comparing the runtime performance with existing methods on the same datasets could provide a clearer picture of its practical applicability.  Can you explain the time overhead? Also, I think the deep learning-based methods are missing in the evaluation. It would be great to involve at least one.
- The paper does not provide sufficient details on how to reproduce the results. e.g. how to get the data points? Is there any preprocess before applying the proposed method? The authors should include a detailed description of the data point extraction process. This would help in understanding the preprocessing steps and their impact on the overall performance of the method. Also, will the source code be released?

Minor questions
- Is there any parameter used in the proposed method? As the proposed method is going through three consensus components, the performance of the proposed method may be sensitive to the selection of model parameters. The authors should provide an analysis of the sensitivity of the method to various parameter settings.
- could you explain more about the motion estimation? As the intro states the proposed method is based on motion estimation but the method seems to combine three types of consensus into a pipeline, I cannot see any functionality of motion-based estimation. Is that about the F, H, A matrix? or what?

**Suitability:**

3

---

### Official Review · Reviewer_2Hjk · 2024-05-26

**Rating:** 4
**Confidence:** 2

**Summary:**

The authors introduce a new method called CMMF, which enhances multi-model fitting by using three diverse consensuses and inter-model collaboration. The method includes a Tangent Consensus Residual Reconstruction (TCRR) module for capturing pixel-level motion structure, a Cross Consensus Affinity (CCA) framework to strengthen correlations between data points and model hypotheses, and a Nested Consensus Clustering (NCC) strategy for handling multi-body motion estimation. Extensive experiments on public datasets demonstrate that CMMF outperforms state-of-the-art methods in terms of accuracy and robustness in the presence of noise and outliers.

**Strengths:**

- In general, this paper is well-structured and articulates the methodology clearly, supported by detailed technical descriptions and illustrative figures (e.g., Fig. 1)
- The experimental results exhibit the promising registration performance of the proposed method across different settings, and present significant performance advantages than other methods.
- The authors offer the sufficient visualization comparisons to support their claims.

**Limitations:**

- The authors should include detailed speed comparisons to provide a fair and comprehensive evaluation.
- The paper lacks comparisons with several highly relevant state-of-the-art multi-model fitting methods, such as [1].

[1] Quantum Multi-Model Fitting, CVPR'2023

**Suitability:**

2

---

### Meta-Review · Area_Chair_nTLj · 2024-07-01

**Recommendation:** Accept (Poster)
**Confidence:** 5

**Metareview:**

All reviewers lean towards acceptance of this paper. The AC concurs with the reviewers and recommends this paper for publication. The authors should address all the issues raised in the reviews in the final version of their paper.